# Determinants of shingles vaccine acceptance in the United Kingdom

**Hélène Bricout[1], Laurence Torcel-Pagnon[1], Coralie Lecomte[2], Mariana F. Almas[3], Ian Matthews[4], Xiaoyan Lu[5]\*, Ana Wheelock[6], Nick Sevdalis[7]**

**1** Sanofi Pasteur, LYON, France, **2** IQVIA, Ouen cedex, France, **3** IQVIA, Berkshire, United Kingdom, **4** MSD Ltd, Maidenhead, United Kingdom, **5** MSD Vaccins, LYON, France, **6** Department of Surgery and Cancer, National Institute for Health Research Imperial Patient Safety Translational Research Centre, Imperial College London, London, United Kingdom, **7** Centre for Implementation Science, King's College London, London, United Kingdom

\* xiaoyan.lu@msd.com

**Data Availability Statement:** All relevant data are within the paper and its Supporting Information files.

**Funding:** This work was funded initially by Sanofi Pasteur MSD and lately by MSD from January

## Abstract

### Background

The United Kingdom (UK) was the first European country to introduce a national immunisation program for shingles (2013–2014). That year, vaccination coverage ranged from 50 to 64% across the UK, but uptake has declined ever since. This study explored determinants of the acceptance of the shingles vaccine in the UK.

### Methods

Vaccinated and unvaccinated individuals, who were eligible for the last catch-up cohort of the 2014–2015 shingles vaccination campaign, were identified using the Clinical Practice Research Datalink (the National Health Service data research service) and invited to participate by their general practitioner (GP). An anonymised self-administered questionnaire was developed using the Health Belief Model as a theoretical framework, to collect data on demographic and socio-economic characteristics, health status, knowledge, influences, experiences and attitudes to shingles and the shingles vaccine. Multivariable logistic regression was used to identify the factors associated with vaccination. Physicians' views concerning perceived barriers to vaccination were also assessed.

### Results

Of the 2,530 questionnaires distributed, 536 were returned (21.2%) from 69 general practices throughout the UK. The majority of responders were female (58%), lived in care homes (56%) and had completed secondary or higher education (88%). There were no differences between vaccinated and unvaccinated responders. Being offered the shingles vaccine by a GP/nurse (odds ratio (OR) = 2.3), and self-efficacy (OR = 1.2) were associated with being vaccinated (p<0.05). In contrast, previous shingles history (OR = 0.4), perceived barriers to vaccination (OR = 0.7) and perceived control of the disease (OR = 0.7) were associated with not being vaccinated against shingles (p<0.05). Less than half (44.0%) of GPs were aware of the local communication campaigns regarding shingles and the shingles vaccine.

2017. Bricout H, Torcel-Pagnon L, Wheelock A and Sevdalis N contributed to the conception/design of the study. Matthews I, Lu X, Lecomte C and Almas MF drafted the initial manuscript. Bricout H and Torcel-Pagnon L were employee of Sanofi Pasteur MSD. Matthews I and Lu X are employed by MDS. Wheelock A and Sevdalis N have received consultancy fees from of Sanofi Pasteur MSD and MDS. Lecomte C and Almas M are employed by IQVIA which received funds from Sanofi Pasteur MSD and MSD to conduct the study. The funder provided support in the form of salaries for authors Bricout H, Torcel-Pagnon L, Matthews I, and Lu X, but did not have any additional role in the study design, data collection of the manuscript. The specific roles of these authors are articulated in the "author contributions" section.

**Competing interests:** Bricout H and Torcel-Pagnon L were employee of Sanofi Pasteur MSD and are currently employed by Sanofi Pasteur. Matthews I and Lu X are employed by MDS. Wheelock A and Sevdalis N have received consultancy fees from of Sanofi Pasteur MSD and MDS. Lecomte C and Almas MF are employed by IQVIA which received funds from Sanofi Pasteur MSD and MSD to conduct the study. Sevdalis N's research is supported by the NIHR Collaboration for Leadership in Applied Health Research and Care South London at King's College Hospital NHS Foundation Trust. Sevdalis N is a member of King's Improvement Science, which is part of the NIHR CLAHRC South London and comprises a specialist team of improvement scientists and senior researchers based at King's College London. Its work is funded by King's Health Partners (Guy's and St Thomas' NHS Foundation Trust, King's College Hospital NHS Foundation Trust, King's College London and South London and Maudsley NHS Foundation Trust), Guy's and St Thomas' Charity, the Maudsley Charity and the Health Foundation. Sevdalis N' research is further supported by an unrestricted research grant by Sanofi Pasteur, as part of an ESRC King's Interdisciplinary Social Sciences Doctoral Training Centre Collaborative Studentship (2017-20). The funders do not have any editorial control over the work reported in this article. The views expressed are those of the author(s) and not necessarily those of King's, the ESRC, the NHS, the NIHR or the Department of Health. This does not alter our adherence to PLOS ONE policies on sharing data and materials.

## Conclusions

Socio-psychological factors largely influence shingles vaccination acceptance in this study. The results add to existing evidence that healthcare providers (HCPs) have a pivotal role against vaccine hesitancy. Campaigns focusing on GPs and accessible information offered to eligible members of the public can further enhance shingles vaccine uptake.

## Introduction

Shingles (herpes zoster) is the clinical manifestation of a reactivation of latent varicella–zoster virus. The incidence of shingles in the United Kingdom (UK) ranges from 3.4–5.0/1,000 person-year and increases to 7.9–8.8/1,000 person-year among those aged 70–79 [1]. Shingles can present several decades after the initial infection with varicella–zoster virus (i.e. varicella), and is characterised by a vesicular skin rash, usually lasting 2 to 4 weeks, often preceded or accompanied by acute pain or itching. About 10–20% of patients with shingles may develop post-herpetic neuralgia (PHN), a debilitating complication where pain persists for more than 3 months [2].

UK was the first European country to introduce shingles vaccination in the 2013–2014 national immunisation programme, targeting adults aged 70 or 79 years (catch-up cohort). For the second year of the programme, in 2014–2015, people aged 78 years on the 1st September 2014 were also targeted for the catch-up programme. The introduction of the shingles vaccine led to about 17,000 fewer episodes of shingles and 3,300 of PHN among 5.5 million eligible individuals in the first 3 years of the programme in England [3]. Vaccination coverage ranged from 50 to 64% across the UK during the first year of the campaign [4–7]. However, uptake has declined in subsequent years [8].

Vaccination hesitancy is a well-recognised obstacle to the success of vaccination programmes [9]. As with any other health-related decision-making process, vaccination behaviour is often influenced by demographic, socio-economic and socio-psychological factors, including beliefs and perceptions towards vaccines [10–13]. Socio-psychological factors are of particular interest as they may be amenable to change.

Few studies have explored factors associated with shingles vaccination [14–18]. Their generalizability remains limited to particular contexts or regions [14,15]. Previous research is also limited by its reliance on self-reported vaccination status [16] and lack of theoretical underpinning model [14–16].

This study sought to address some of these shortcomings. We aimed to explore, for the first time to our knowledge, the constellation of factors which may influence shingles vaccine uptake in the UK. To this end, we employed a theory-driven framework for attitudinal assessment, the Health Belief Model (HBM). The HBM has been widely used to study health-seeking behaviours including vaccine acceptance in the elderly, mainly influenza vaccine [17–19], but also the shingles vaccine [20,21]. As a secondary aim, GPs' views concerning barriers to shingles vaccination were also assessed.

## Methods

### Sampling strategy

The Clinical Practice Research Datalink (CPRD), the UK governmental data research service based on anonymised primary care records, was used to identify individuals vaccinated and

unvaccinated against shingles among those eligible for the last catch-up cohort of the 2014–2015 vaccination campaign (aged 79), thus not eligible for the following campaign. All individuals born in 1934 and 1935 were mapped to their practices. Eligible individuals were sent an anonymous self-administered paper questionnaire from their GP's practice.

Our sample size calculation is based on the estimation that a sample size of 500 patients (1:1 vaccinated versus unvaccinated) could detect an odds ratio (OR) ranging between 1.66–2.08 (two-sided $\alpha = 5\%$, $\beta = 80\%$). This is consistent with the ORs observed in a study investigating the impact of shingles vaccine awareness on immunisation among people aged $\geq 50$ years [22]. A response rate of 20% was expected as elderly individuals are less likely to return completed postal questionnaires [23]. Thus 2,500 individuals were targeted.

Ninety-one practices with $\geq 30$ individuals in each birth cohort were selected based on their interest to participate, geographic dispersion across UK (England, Wales, Scotland, and Northern Ireland), practice size and research experience. They were provided with a list of potentially eligible individuals based on their year of birth who were randomly selected (up to 60 per practice). Study responder characteristics, including vaccination status (confirmed by the GPs in the primary care records transferred to CPRD anonymously), were assessed after 100 and 300 questionnaires were received, to monitor any ongoing selection bias (to get closer to a 1:1 ratio of individuals vaccinated and unvaccinated against shingles).

## Data collection

The HBM underpinned the development of the attitudinal assessment instrument. We assessed perceived susceptibility, severity, barriers, cue to action and self-efficacy (i.e. confidence in one's ability to take action) [24,25] in relation to shingles and the shingles vaccine. The instrument also measured socio-demographic variables and was informed by recent evidence on behavioural factors that affect vaccination uptake [19]. Further, health decision-making preferences [26,27], knowledge [12], perceived control of the disease [28,29] and trust in key vaccination stakeholders [30] were also investigated based on prior evidence of the relevance of these factors on vaccination uptake. After concept elaboration, cultural and semantic review, conceptual equivalence check and independent proofreading, the survey instrument was pilot tested in 5 healthy adults eligible for the shingles vaccination campaign recruited from UK community centres to ensure feasibility and comprehension. The participants were asked to complete the questionnaire, to comment on the response options and on items difficult to understand, suggesting alternative wording, followed by cognitive debriefing. Afterwards there was another round of instrument developer review and final proofreading.

Objective vaccination data collected included vaccination status, gender and year of birth, retrieved directly from the CPRD.

GPs' views about shingles vaccination were also assessed. A paper survey was sent to each GP practice, which assessed vaccination practices, local communication campaigns on shingles vaccination and perceived barriers to shingles vaccination.

## Data analysis

The survey items were answered on multiple or alternate choice and 7-point Likert scales [31,32]. All analyses were performed using SAS. Descriptive statistics were produced for all survey responses. Items reflecting HBM constructs were aggregated into the relevant composite constructs, where internal consistency was considered satisfactory if Cronbach's alpha coefficient was $\geq 0.70$ [33]. Bivariate analysis (chi-squared and t-tests) compared responders versus non-responders' socio-demographic characteristics and vaccination status, and vaccinated versus unvaccinated responder's answers to the survey (two-sided tests, $\alpha = 5\%$).

A multivariable logistic regression model was produced using HBM constructs and other socio-psychological factors, as well as socio-demographic and health factors, as independent variables; and objective vaccination status as the dependent variable.

Complete case analysis was used for the multivariable model. Robustness was assessed with sensitivity analyses, assuming an arbitrary missing pattern using Markov chain Monte Carlo. Multiple imputation for all Likert-scale items with missing data used the established procedure by Rubin et al [34]. GP survey items were analysed descriptively.

### Ethics review

The study protocol was approved by the National Health Service Research Ethics Committee (reference number: 15/SC/0503), the National Research Ethics Service, the local NHS trust of the practices and the Independent Scientific Advisory Committee for CPRD access. A participant information sheet was provided to the individuals with the survey. An individual's decision to complete and return the survey was interpreted as consent to participate.

## Results

### Responders characteristics

From the 91 contacted GP practices, 84 (92.3%) accepted to participate. Among 2,530 questionnaires distributed by those practices, 536 were returned (21.2%) from 69 practices throughout the UK. There were relatively fewer responders from England and more from Northern Ireland and Scotland. The shingles vaccine coverage was 70.1% among responders and 58.9% among non-responders (Table 1).

Most responders were female (57.8%), of white ethnicity (98.8%), belonged to urban practices (61.8%), living in a care home (55.7%), had completed high/secondary school or higher

**Table 1. Responders and non-responders characteristics.**

|  | All Individuals (N = 2530) | Responders (N = 536) | Non-Responders (N = 1994) | p-value |
|---|---|---|---|---|
| **Sex** |  |  |  | 0.344 |
| Male | 1087 (43.0%) | 226 (42.2%) | 861 (43.2%) |  |
| Female | 1443 (57.0%) | 310 (57.8%) | 1133 (56.8%) |  |
| Missing | 0 | 0 | 0 |  |
| **Shingles vaccination status** |  |  |  | 0.123 |
| Vaccinated | 1318 (61.4%) | 344 (70.1%) | 974 (58.9%) |  |
| Unvaccinated | 828 (38.6%) | 147 (29.9%) | 681 (41.1%) |  |
| Missing | 384 | 45 | 339 |  |
| **Nation** |  |  |  | <0.001 |
| England | 1515 (59.9%) | 273 (50.9%) | 1242 (62.3%) |  |
| Northern Ireland | 439 (17.4%) | 118 (22.0%) | 321 (16.1%) |  |
| Scotland | 160 (6.3%) | 53 (9.9%) | 107 (5.4%) |  |
| Wales | 416 (16.4%) | 92 (17.2%) | 324 (16.2%) |  |
| Missing | 0 | 0 | 0 |  |
| **GP's geographical location** |  |  |  | <0.001 |
| Urban | 1709 (75.3%) | 289 (61.8%) | 1420 (78.8%) |  |
| Rural | 561 (24.7%) | 179 (38.2%) | 382 (21.2%) |  |
| Missing | 260 | 68 | 192 |  |

p-value for Chi-Square test (two-sided)

(88.4%), and were not engaged in any professional/caring activity (71.8%). Only half of responders reported their income. Many responders (77.0%) considered themselves to be in good health. Nonetheless, two-thirds of responders reported having at least 1 medical condition; diabetes being the most common. Unvaccinated responders presented a higher prevalence of diabetes and history of shingles in the past than vaccinated responders (S1 Table).

## Knowledge and perceptions of shingles and the shingles vaccine

The average self-assessed knowledge about shingles by the responders was 3.4 on the 7-point scale. True knowledge about shingles was further assessed by 4 statements (Fig 1). Regardless of vaccination status, most of the responders answered correctly, thus exhibiting accurate knowledge. However, approximately one-third of participants did not know that shingles could not be caught from another person with shingles, nor that the chance of developing shingles increases with age.

There were few significant differences on perceptions of shingles and the shingles vaccine between vaccinated and unvaccinated responders (p<0.05) as summarised in Table 2 (complete list of studied determinants is displayed in S1 Table).

Vaccinated responders perceived lower susceptibility to shingles, scored slightly higher on the perceived benefits of the shingles vaccine and on vaccine related self-efficacy, and felt less constrained by the practical barriers as compared with unvaccinated. In contrast, unvaccinated responders perceived more barriers to the shingles vaccine and had a higher perceived control of the disease without the vaccine.

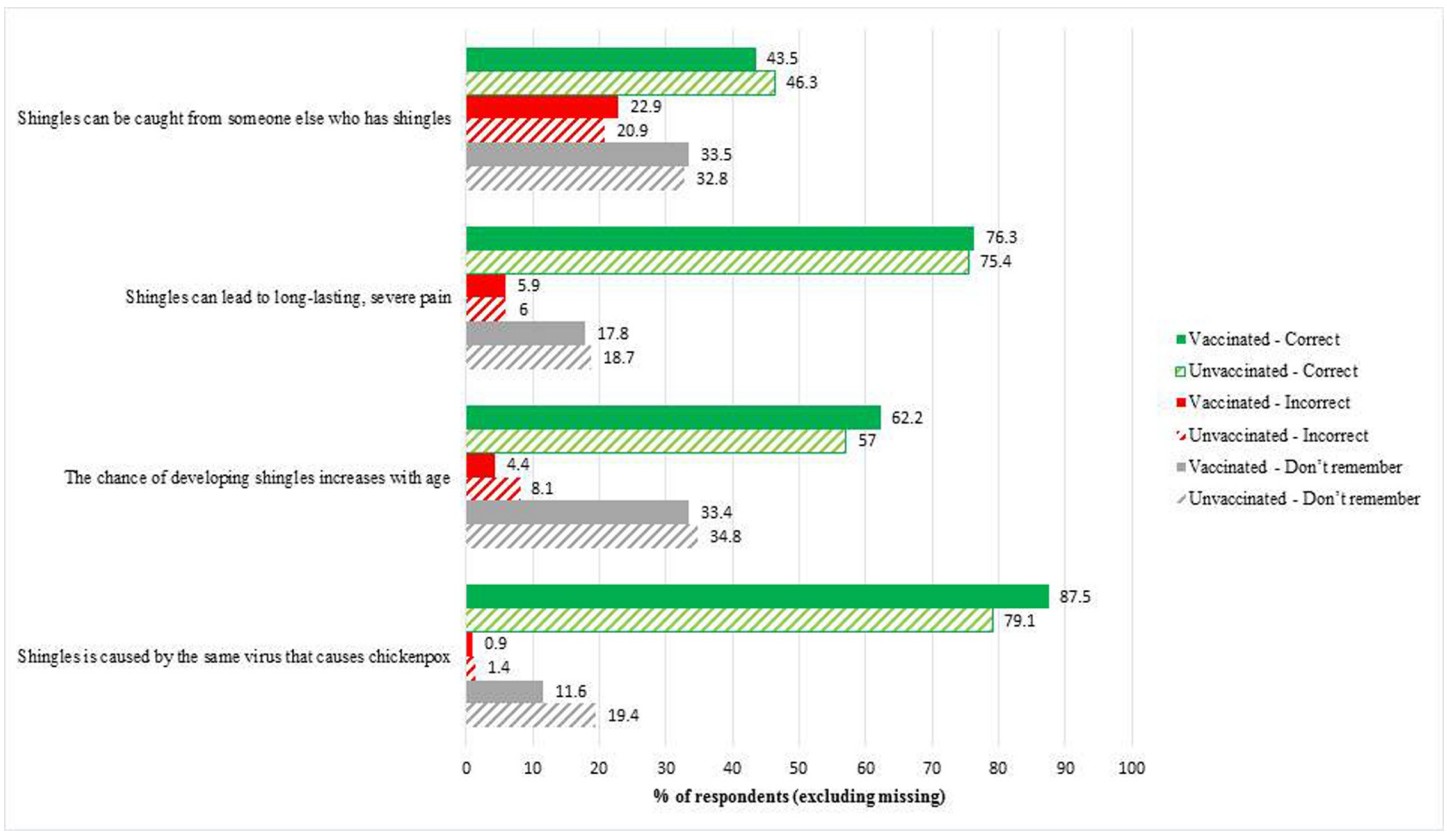

**Fig 1. True knowledge about shingles among responders.** Note: "Shingles can be caught from someone else who has shingles." (Correct answer: false); "Shingles can lead to long-lasting, severe pain." (Correct answer: true); "The chance of developing shingles increases with age." (Correct answer: true); "Shingles is caused by the same virus that causes chickenpox." (Correct answer: true).

**Table 2. Main determinants of shingles vaccination.**

| | Bivariate Analysis (N = 501) | | | | | Multivariable Model (N = 348) | |
|---|---|---|---|---|---|---|---|
| | Vaccinated (N = 344) | | Unvaccinated (N = 147) | | p-value | OR | 95% CI |
| | n (%) | Mean (SD) | n (%) | Mean (SD) | | | |
| **History of shingles** | 344 | | 145 | | | | |
| *No[1]* | 239 (69.5) | - | 81 (55.9) | - | <0.001 | 1.0[1] | |
| *I don't know/remember[1]* | 17 (4.9) | - | 3 (2.1) | - | | | |
| *Yes* | 88 (25.6) | - | 61 (42.1) | - | | 0.4*γ | 0.2–0.7 |
| **Perceived susceptibility** | 332 | 3.6 (1.27) | 142 | 3.8 (1.24) | 0.036 | 1.0 | 0.8–1.3 |
| **Perceived benefits** | 327 | 5.6 (1.11) | 136 | 5.3 (1.03) | 0.002 | 1.0 | 0.8–1.4 |
| **Perceived barriers** | 332 | 2.9 (1.08) | 138 | 3.6 (0.99) | <0.001 | 0.7 | 0.5–1.0 |
| **Practical barriers and Facilitators** | 329 | 6.2 (1.40) | 129 | 5.8 (1.62) | 0.008 | 1.0 | 0.8–1.3 |
| **Self-efficacy** | 316 | 5.9 (1.80) | 127 | 5.2 (2.16) | <0.001 | 1.2*γ | 1.0–1.4 |
| **Perceived control of disease** | 323 | 2.7 (1.56) | 130 | 3.5 (1.54) | <0.001 | 0.7*γ | 0.6–0.9 |
| **Did your GP or nurse offer you the shingles vaccination (through a letter, phone call, text message or during a visit)?** | 321 | | 139 | | | | |
| *I don't know/remember[1]* | 35 (10.9%) | - | 16 (11.5%) | - | | 1.0[1] | |
| *No[1]* | 66 (20.6%) | - | 56 (40.3%) | - | <0.001 | | |
| *Yes* | 220 (68.5%) | - | 67 (48.2%) | - | | 2.3*γ | 1.1–4.7 |
| **Did your GP or nurse tell you about shingles?** | 315 | | 131 | | | | |
| *I don't know/remember[1]* | 47 (14.9%) | - | 13 (9.9%) | - | | 1.0[1] | |
| *No[1]* | 126 (40.0%) | - | 79 (60.3%) | - | <0.001 | | |
| *Yes* | 142 (45.1%) | - | 39 (29.8%) | - | | 0.7 | 0.3–1.4 |
| **Do you know anyone who has had the shingles vaccination?** | 325 | | 137 | | | | |
| *I don't know/remember[1]* | 19 (5.8%) | - | 5 (3.6%) | - | | 1.0[1] | |
| *No[1]* | 151 (46.5%) | - | 92 (67.2%) | - | <0.001 | | |
| *Yes* | 155 (47.7%) | - | 40 (29.2%) | - | | 1.6 | 0.8–3.2 |
| **Did anyone, among your vaccinated relatives or friends, advise you to have the shingles vaccination?** | 325 | | 135 | | | | |
| *I don't know/remember[1]* | 29 (8.9%) | - | 6 (4.4%) | - | | 1.0[1] | |
| *No[1]* | 242 (74.5%) | - | 117 (86.7%) | - | 0.016 | | |
| *Yes* | 54 (16.6%) | - | 12 (8.9%) | - | | 1.6 | 0.6–4.4 |
| Max-rescaled R-Square (pseudo-$R^2$) | | | | | | 0.3220 | |

CI = Confidence Interval; OR = Odds ratio;

*p ≤ 0.05;

γ direction and significance of effect corroborated in sensitivity analysis.

[1] Multivariable model reference category is "Other than yes". It includes "No", "I don't know/remember" and missing.

Note: the complete list of studied determinants is displayed in S1 Table.

Responders were generally engaged with GPs in medical decision-making (>85%) and highly trusted their GP and the NHS recommendations regarding shingles. For most responders, information about the shingles vaccine was obtained whilst they were attending the doctor's surgery (75.9% among vaccinated and 67.3% among unvaccinated, p = 0.05). Vaccinated responders were more likely than the unvaccinated ones to have been offered the shingles vaccine by their GP/nurse, told about shingles by their GP/nurse, advised to receive the vaccine by vaccinated relatives or friends, or know someone who had shingles vaccination.

## Determinants of shingles vaccination uptake

As summarized in Table 2, shingles vaccination was associated with GP/nurse vaccine recommendations (OR: 2.3; 95% Confidence Interval (CI): 1.1–4.7; p<0.05) and vaccine related self-efficacy construct (OR: 1.2; 95% CI: 1.0–1.4; p<0.05). In contrast, non-vaccination was associated with perceived barriers (OR: 0.7; 95% CI: 0.5–1.0; p<0.05), perceived control of the disease (OR: 0.7; 95% CI: 0.6–0.9; p<0.05) and previous history of shingles (OR: 0.4, 95% CI: 0.2–0.7; p<0.05). Approximately one-third of the observations had at least 1 missing variable and therefore were excluded from the model. Encouragingly, the sensitivity analyses using multiple imputation corroborated the results from the multivariable model presented in S1 Table.

## GPs' perceptions regarding shingles vaccination in their practices

The majority of GPs considered that their practices had internal procedures/guidelines (95.1%) and enough staff (90.5%) to provide vaccination information to the elderly and had materials available for patients (91.7%). Most GPs considered having enough time to provide vaccination recommendations to their elderly patients (72.7% of those in rural and 57.6% of those in urban settings). Approximately one-third of GPs either stated that there were no local communication campaigns (e.g. local radio/TV spot, local newspapers advertisement, etc.) regarding shingles vaccination or preferred not to answer.

Responses from rural and urban GPs on their opinion about shingles and the shingles vaccine were comparable (Fig 2). GPs had a neutral opinion or slightly agreed that shingles was an economic burden, the shingles vaccine was effective, there was enough information on the duration of protection of the shingles vaccine, that their patients thought they needed the vaccine or were concerned with getting the shingles vaccine. The single injection for the shingles vaccine was strongly seen as an advantage by the GPs.

## Discussion

To the best of our knowledge, this is the first study conducted in the UK investigating the determinants of the acceptance of the shingles vaccine using a theory-informed instrument and objectively derived vaccination status. The vaccinated responders were more likely to have been offered the shingles vaccine by their GP/nurse or advised to take it by their relatives or friends. They also reported feeling less susceptible to shingles, were more likely to value the benefits of the shingles vaccine, scored higher on perceived vaccine related self-efficacy and were significantly less constrained by practical barriers to vaccination. In contrast, unvaccinated responders were more likely to report practical barriers to shingles vaccination, and believed they were more able to control the disease without the vaccine. Our regression model accounted for one-third of the variability in the shingles vaccination uptake in our sample.

Our results support findings from previous studies and add new insights. Self-reported knowledge about shingles was limited and consistent with the results of a global survey where little or no knowledge of shingles was reported across regions [35]. Although, the majority of UK responders knew shingles is caused by the same virus that causes chickenpox, the majority did not know or did not remember that shingles cannot be transmitted from another person with shingles. In addition, 87.1% of the responders had limited knowledge on shingles vaccine, consistent with existing literature [16]. Our findings are discrepant, however, with those from a global survey suggesting that responders with prior experience of shingles were more likely to be aware of shingles and believed they could develop it, indicating they would be more likely get the vaccine [35]. Data from the US 2007 National Immunization Survey-Adult (NIS-Adult) indicated that one of the main reasons for not accepting the shingles vaccine was participants felt vaccination was not needed (34.8%) [36]. Responders who had the disease in the past may

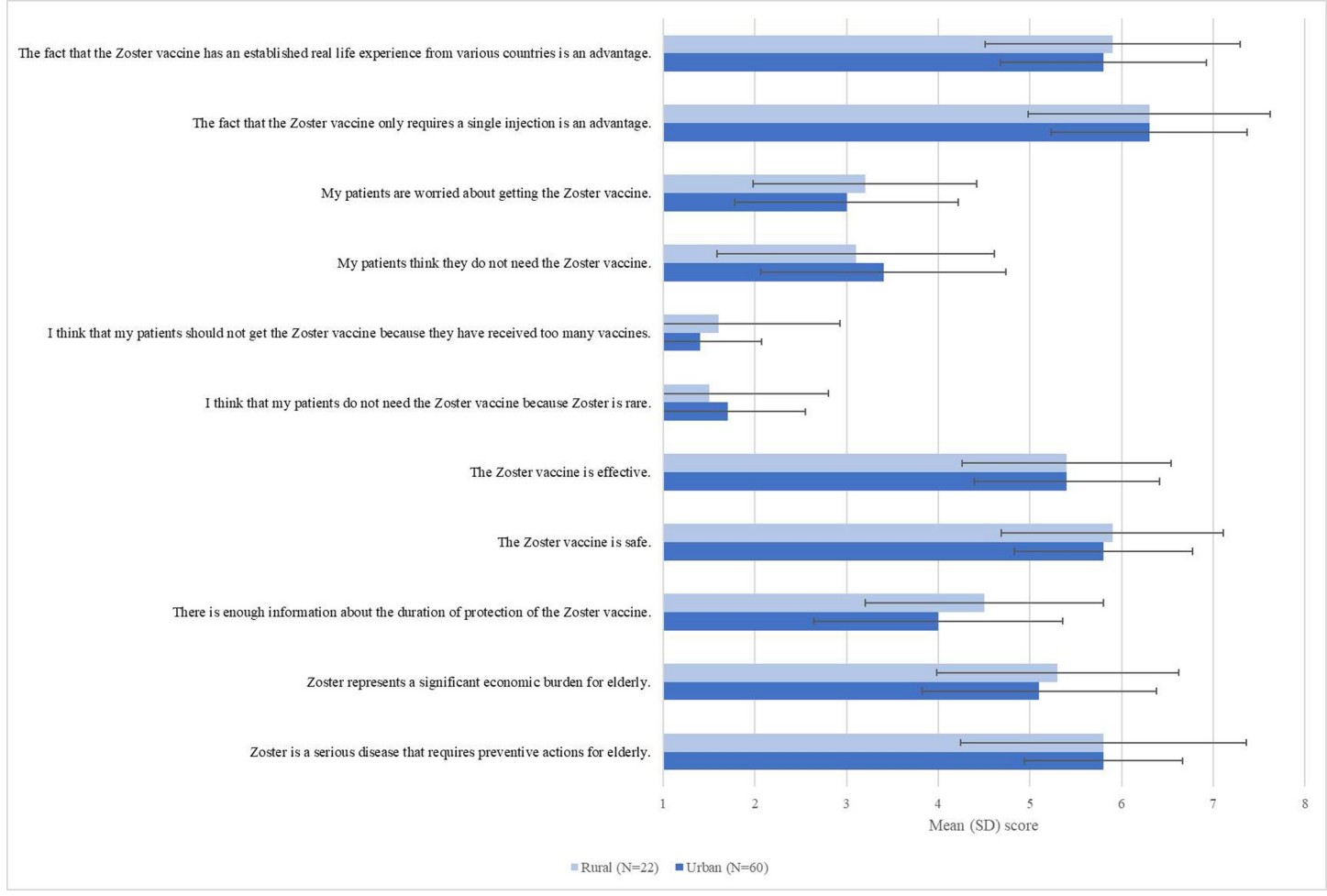

**Fig 2. GPs' perceptions of shingles and the shingles vaccine (1 = strongly disagree, 7 = strongly agree).**

consider that they do not need the shingles vaccine, either because of increased awareness to acquired boosted immunity after the first episode of shingles, or because they managed the previous episode without it. Since the incidence of recurrent shingles and the relationship with previous episodes are still under investigation [37], it is possible that GPs and individuals consider that the vaccine is not required if the disease occurred in the past.

The study adds to evidence on the pivotal role of provider recommendations regarding getting the shingles vaccine. Consistent with other studies, receiving advice from a GP or another healthcare provider (HCP) to get the shingles vaccine increases acceptability of the vaccine [15,16,21,38,39], and may even reverse initial reluctance towards the shingles vaccination [14].

Overall, our results highlight the importance of routine monitoring and addressing vaccination sentiment among cohorts eligible for shingles vaccination–as this can offer useful insights regarding objective uptake of vaccination.

## Limitations and strengths

The study targeted larger GP practices more familiar with research and where ethical approval procedures were streamlined. Due to delays in the study implementation, there was an over-representation of Northern Ireland and underrepresentation of England and Scotland–hence

overall the study is not representative of the UK eligible population for the shingles vaccine. Also, practices were mostly located in the urban areas. People living in a care home or assisted accommodation were overrepresented when compared with an estimate of only 3% in 2011 among 75–84 years old UK residents [40]. This may have led to recall bias, thus explaining the high proportion for some items answered as "I don't know/remember"; however, this bias could not be quantified. Our study was not designed to assess the role of ethnicity as a determinant of shingles vaccine uptake; yet a recent publication has suggested plays a role in shingles vaccine uptake [41]. This aspect requires further detailed investigation. The proportion of white participants in our sample was in line with census data for elderly patients (99.8% white people in Northern Ireland, 95.1% in England and Wales and 99.2% in Scotland) [42–44]. Our achieved response rate was approximately 20% which is low but is in line with what was expected for such a study and consistent with earlier similar research [23]. A larger proportion of responders were vaccinated (70.1%) compared to the initial target 50%±10% which may indicate participation bias as individuals responding to the survey were more compliant with vaccination. it should be noted that only the older catch-up cohort for shingles vaccination campaign was assessed, to avoid a possible influence on the vaccination behaviour after the study among the participants. Therefore, this limits the ability to generalise the results to a younger population in a context of a different immunisation programme.

The study also has methodological strengths. The assessed variables were based on a well-established conceptual model (HBM) and recent research on socio-psychological vaccination determinants. GP practices were selected from CPRD, which is considered representative of GP practices throughout the UK, and the number of individuals per practice was capped to avoid cluster effects. The regression model generated in this study comprised a comprehensive set of variables, which have been previously associated with preventive health-seeking behaviour. The objective assessment of vaccination status using an electronic database rather than self-reported data addresses a key limitation of many similar studies in the field.

### Policy and practice recommendations

To improve individuals' knowledge about shingles, the messages conveyed to the public should emphasise the cause of shingles, how it is triggered and the possible complications, such as PHN. In addition, current results suggest that people who had shingles in the past may not know the vaccine can protect them from future episodes. Whilst future research is required to further explore this hypothesis, current campaigns should encourage the uptake of the vaccine among those who experienced shingles in the past. The results of this study suggest that despite both vaccinated and unvaccinated responders learned about the shingles vaccine at the GP practice, being offered the vaccine directly by a HCP seems to be key in their decision-making process. Therefore, vaccination campaigns should focus on the HCPs, given the importance of the recommendations of these professionals on vaccination uptake.

We also found that less than half of the GPs surveyed were aware of local communication campaigns regarding the shingles vaccine, but the majority considered their practices had internal procedures/guidelines, materials and sufficient staff to provide vaccine information. These results indicate that knowledge about the vaccine at GP practice level can be improved notably on the economic burden of shingles to society, the duration of protection of the shingles vaccine and vaccine effectiveness confirmed by recent findings conducted on the 3 first years of the UK vaccination programme [3,45,46]. Communication campaigns should emphasise to HCPs the relevance of engaging with their patients to understand their motivation and concerns regarding the shingles vaccine as an important lever to improve vaccination coverage.

## Conclusion

The UK's national immunisation program to prevent shingles has proved successful in preventing this debilitating condition, but its benefits are dependent on the uptake of the shingles vaccine. Our study suggests that policy amenable socio-psychological factors can explain the likelihood of vaccination uptake for this condition better than socio-demographic factors alone. Being proactively offered the shingles vaccine by a GP or a nurse, perceiving to be at risk of developing shingles and perceived self-efficacy are associated with shingles vaccination uptake. Our results further add to the existing evidence that HCPs have a pivotal role in promoting herpes zoster vaccination. Future campaigns should focus on GPs and offer eligible members of the public accessible information regarding shingles to further promote vaccination uptake.

## Supporting information

**S1 Table. Determinants of shingles vaccination.**
(DOCX)

**S1 Appendix. Individual questionnaire.**
(DOCX)

**S2 Appendix. GP questionnaire.**
(DOCX)

## Acknowledgments

Temt D and Ngo T from IQVIA have participated in the conduct and analysis of this study, respectively. Boyle E from the CPRD, Bertrand I from (formerly) Sanofi Pasteur MSD and Engel P from IQVIA, on top of the authors, were part of the study steering committee. Bertrand I, Thomas S, Dard S and Bosc P (former employees of Sanofi Pasteur MSD) have contributed to the design of the study and study set up. Boyle E has coordinated the set up and conduct of the study at CPRD. We thank Johnson K and Zhang D (Merck & Co., Inc.) for providing technical assistance, Samant S for study management of the study and manuscript, and Wright M (Head of Real-World Clinical Studies, CPRD) for providing project oversight.

## Author Contributions

**Conceptualization:** Hélène Bricout, Laurence Torcel-Pagnon, Ana Wheelock, Nick Sevdalis.

**Data curation:** Coralie Lecomte.

**Formal analysis:** Coralie Lecomte.

**Funding acquisition:** Hélène Bricout, Xiaoyan Lu.

**Investigation:** Hélène Bricout, Xiaoyan Lu, Ana Wheelock, Nick Sevdalis.

**Methodology:** Hélène Bricout, Laurence Torcel-Pagnon, Coralie Lecomte, Mariana F. Almas, Ana Wheelock, Nick Sevdalis.

**Project administration:** Laurence Torcel-Pagnon, Mariana F. Almas, Xiaoyan Lu.

**Resources:** Laurence Torcel-Pagnon.

**Supervision:** Coralie Lecomte, Ian Matthews, Xiaoyan Lu, Ana Wheelock, Nick Sevdalis.

**Validation:** Hélène Bricout, Ian Matthews, Xiaoyan Lu, Ana Wheelock, Nick Sevdalis.

**Writing – original draft:** Coralie Lecomte, Mariana F. Almas.

**Writing – review & editing:** Hélène Bricout, Laurence Torcel-Pagnon, Coralie Lecomte, Mariana F. Almas, Ian Matthews, Xiaoyan Lu, Ana Wheelock, Nick Sevdalis.

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
