## [Decision Letter · Decision Letter 0]

18 Jun 2019

PONE-D-19-14771

Determinants of shingles vaccine acceptance in the United Kingdom

PLOS ONE

Dear Dr. Lu,

Thank you for submitting your manuscript to PLOS ONE. After careful consideration, we feel that it has merit but does not fully meet PLOS ONE’s publication criteria as it currently stands. Therefore, we invite you to submit a revised version of the manuscript that addresses the points raised during the review process.

We would appreciate receiving your revised manuscript by July 15, 2019. To enhance the reproducibility of your results, we recommend that if applicable you deposit your laboratory protocols in protocols.io, where a protocol can be assigned its own identifier (DOI) such that it can be cited independently in the future. For instructions see: http://journals.plos.org/plosone/s/submission-guidelines#loc-laboratory-protocols

We look forward to receiving your revised manuscript.

Kind regards,

Italo Francesco Angelillo, DDS, MPH

Academic Editor

PLOS ONE

Journal Requirements:

3. Please upload a copy of Figure 1, to which you refer in your text on page 5. If the figure is no longer to be included as part of the submission please remove all reference to it within the text.

Reviewers' comments:

Reviewer's Responses to Questions

**Comments to the Author**

1. Is the manuscript technically sound, and do the data support the conclusions?

Reviewer #1: Yes

Reviewer #2: Yes

2. Has the statistical analysis been performed appropriately and rigorously? 

Reviewer #1: Yes

Reviewer #2: Yes

3. Have the authors made all data underlying the findings in their manuscript fully available?

Reviewer #1: Yes

Reviewer #2: Yes

4. Is the manuscript presented in an intelligible fashion and written in standard English?

Reviewer #1: Yes

Reviewer #2: Yes

5. Review Comments to the Author

Reviewer #1: This revised version of the paper addresses all my previous points.

I do not have any further comment to do.

Reviewer #2: The article provides interesting information on the determinants of shingles vaccination uptake in a sample of UK elders, identified through the Clinical Practice Research Datalink (CPRD). As also the authors stated, is one of the few studies that have been conducted to assess the determinants of vaccine hesitancy specifically addressed at shingles vaccination. The methodology adopted is sound and well described in the manuscript.

I would recommend describing in the methods section the items used for assessing Perceived susceptibility, Perceived benefits, Perceived barriers, Practical barriers and Facilitators, Self-efficacy and Perceived control of disease, or alternatively use a footnote to Table 2, to help the reader understand the underlying elements of the HBM theoretical construct.

In the Discussion section, the authors state that “The study adds to evidence on the pivotal role of provider recommendations regarding vaccines in general and specifically getting the shingle vaccine” (lines 181-182). I would suggest also citing some research conducted on the acceptance of varicella vaccination in pediatric age, particularly in Italy, where the vaccination against varicella became mandatory in 2017 (e.g. Vezzosi L, Santagati G, Angelillo IF. Knowledge, attitudes, and behaviors of parents towards varicella and its vaccination. BMC Infect Dis. 2017 Feb 27;17(1):172. doi: 10.1186/s12879-017-2247-6; Rosso A, Massimi A, De Vito C, Adamo G, Baccolini V, Marzuillo C, Vacchio MR, Villari P. Knowledge and attitudes on pediatric vaccinations and intention to vaccinate in a sample of pregnant women from the City of Rome.Vaccine. 2019 Mar 28;37(14):1954-1963. doi: 10.1016/j.vaccine.2019.02.049; but also the German study Hagemann C, Streng A, Kraemer A, Liese JG. Heterogeneity in coverage for measles and varicella vaccination in toddlers - analysis of factors influencing parental acceptance. BMC Public Health. 2017 Sep 19;17(1):724. doi: 10.1186/s12889-017-4725-6.). All these studies identified the information received from healthcare providers as one of the determinants of vaccine acceptance for a disease which severity tends to be underestimated.

6. PLOS authors have the option to publish the peer review history of their article (what does this mean?). If published, this will include your full peer review and any attached files.

Reviewer #1: Yes: Giovanni Gabutti

Reviewer #2: No

---

## [Author Response · Author response to Decision Letter 0]

9 Jul 2019

Reviewer #1: This revised version of the paper addresses all my previous points.

I do not have any further comment to do.

Authors’ response: Thank you.

Reviewer #2: The article provides interesting information on the determinants of shingles vaccination uptake in a sample of UK elders, identified through the Clinical Practice Research Datalink (CPRD). As also the authors stated, is one of the few studies that have been conducted to assess the determinants of vaccine hesitancy specifically addressed at shingles vaccination. The methodology adopted is sound and well described in the manuscript.

I would recommend describing in the methods section the items used for assessing Perceived susceptibility, Perceived benefits, Perceived barriers, Practical barriers and Facilitators, Self-efficacy and Perceived control of disease, or alternatively use a footnote to Table 2, to help the reader understand the underlying elements of the HBM theoretical construct.

In the Discussion section, the authors state that “The study adds to evidence on the pivotal role of provider recommendations regarding vaccines in general and specifically getting the shingle vaccine” (lines 181-182). I would suggest also citing some research conducted on the acceptance of varicella vaccination in pediatric age, particularly in Italy, where the vaccination against varicella became mandatory in 2017 (e.g. Vezzosi L, Santagati G, Angelillo IF. Knowledge, attitudes, and behaviors of parents towards varicella and its vaccination. BMC Infect Dis. 2017 Feb 27;17(1):172. doi: 10.1186/s12879-017-2247-6; Rosso A, Massimi A, De Vito C, Adamo G, Baccolini V, Marzuillo C, Vacchio MR, Villari P. Knowledge and attitudes on pediatric vaccinations and intention to vaccinate in a sample of pregnant women from the City of Rome.Vaccine. 2019 Mar 28;37(14):1954-1963. doi: 10.1016/j.vaccine.2019.02.049; but also the German study Hagemann C, Streng A, Kraemer A, Liese JG. Heterogeneity in coverage for measles and varicella vaccination in toddlers - analysis of factors influencing parental acceptance. BMC Public Health. 2017 Sep 19;17(1):724. doi: 10.1186/s12889-017-4725-6.). All these studies identified the information received from healthcare providers as one of the determinants of vaccine acceptance for a disease which severity tends to be underestimated.

Authors’ response: Thank you for your suggestions. 

We have incorporated a footnote to Table 2 to refer to the detailed table in Supporting Information (S1 Table) where items used for each individual construct are displayed.

This study was to assess the determinants of shingles vaccination as part of a national immunisation programme in the older population. Consequently, the authors have selected most relevant papers related to the study population and vaccine considered. We have carefully considered the research suggested however they refer to paediatric population/parental acceptance or pregnant women which were not the main population of interest in our research; thus, they were not added to the manuscript. The statement in the discussion section was revised as follows “The study adds to evidence on the pivotal role of provider recommendations regarding getting the shingle vaccine” as specific references related to the shingles vaccine were already included in the manuscript in the next sentence.

---

## [Editor Report · Decision Letter 1]

12 Jul 2019

Determinants of shingles vaccine acceptance in the United Kingdom

PONE-D-19-14771R1

Dear Dr. Lu,

We are pleased to inform you that your manuscript has been judged scientifically suitable for publication and will be formally accepted for publication once it complies with all outstanding technical requirements.

With kind regards,

Italo Francesco Angelillo, DDS, MPH

Academic Editor

PLOS ONE
---

## [Editor Report · Acceptance letter]

19 Jul 2019

PONE-D-19-14771R1 

Determinants of shingles vaccine acceptance in the United Kingdom 

Dear Dr. Lu:

I am pleased to inform you that your manuscript has been deemed suitable for publication in PLOS ONE. Congratulations! Your manuscript is now with our production department. 

With kind regards,

on behalf of

Professor Italo Francesco Angelillo 

Academic Editor

PLOS ONE